

# Survival, growth and carbon content in a forest plantation established after a clear-cutting in Durango, Mexico

Jesús Alejandro Soto-Cervantes[1], Artemio Carrillo-Parra[2],
Rodrigo Rodríguez-Laguna[3], José Javier Corral-Rivas[4],
Marín Pompa-García[4] and Pedro Antonio Dominguez-Calleros[4]

[1] Programa Institucional de Doctorado en Ciencias Agropecuarias y Forestales, Universidad Juárez del Estado de Durango, Durango, Mexico
[2] Instituto de Silvicultura e Industria de la Madera, Universidad Juárez del Estado de Durango, Durango, Mexico
[3] Instituto de Ciencias Agropecuarias, Universidad Autónoma del Estado de Hidalgo, Tulancingo, Hidalgo, Mexico
[4] Facultad de Ciencias Forestales, Universidad Juárez del Estado de Durango, Durango, México

Corresponding author
Pedro Antonio Dominguez-Calleros,
pdomingc@hotmail.com

## ABSTRACT

**Background**. Forest plantations play an important role in carbon sequestration, helping to mitigate climate change. In this study, survival, biomass, growth rings and annual carbon content storage were evaluated in a mixed *Pinus durangensis* and *P. cooperi* plantation that was established after a clear-cutting. The plantation is eight years old and covers an area of 21.40 ha.

**Methods**. Sixteen sites of 100 m$^2$ were distributed randomly. At each site, two trees distributed proportionally to the diametric categories were destructively sampled (one per tree species). Two cross-sections were cut from each tree: The first at the base of the stump and the second at 1.30 m. The width of tree rings of the first cross-section was measured using a stereoscopic microscope with precision in microns ($\mu$m). The year-by-year basal diameter of each tree was recorded and biomass and carbon content was estimated using allometric equations.

**Results**. The estimated survival was 75.2%. The results of the ANOVA showed significant differences between the year-by-year width records of tree rings, the highest value corresponding to the fifth year. The average carbon sequestration per year is 0.30 kg for both studied tree species.

**Conclusions**. *P. durangensis* and *P. cooperi* plantations adapt and develop well in Durango forests when they are established in areas that are subjected to clear-cutting.

# INTRODUCTION

In addition to the multiple benefits that forest ecosystems provide to society, forests can capture significant amounts of greenhouse gases (GHG), particularly carbon dioxide (*Benjamín & Masera, 2001*; *Aguirre-Calderón & Jiménez-Pérez, 2011*; *Martínez et al., 2016*; *González-Cásares et al., 2019*). However, the vegetation cover is not always adequate due to

inappropriate forest management, and so the establishment of trees is necessary. Whatever the purpose of this action, reforestation (forest plantations) is an excellent alternative to mitigate high atmospheric concentrations of $CO_2$ and, at the same time, reducing global warming (*Van Minnen et al., 2008*; *López-Reyes, 2016*; *Patiño et al., 2018*; *Ramírez-López & Chagna-Avila, 2019*). Proper soil management also contributes significantly to the expansion of the carbon sink in the terrestrial biosphere (*Zambrano, Franquis & Infante, 2004*; *Caviglia, Wingeyer & Novelli, 2016*; *Halifa-Marín et al., 2019*).

The Mexican forests have been considered as diverse in terms of tree species (*Medrano et al., 2017*) and have great potential as a carbon sink, and therefore are considered essential to assess carbon content in programs designed to mitigate global warming (*Pompa-García & Sigala-Rodríguez, 2017*; *Domínguez-Calleros et al., 2017*). Frequently 50% carbon concentration of total biomass has been assumed, however, this statement leads to inaccurate carbon estimates, due to variation of the carbon concentration between arboreal components and tree species (*Pompa-García & Yerena-Yamalliel, 2014*; *Wang et al., 2015*; *Hernández-Vera et al., 2017*). Forest plantations of one or two species stand out as the most efficient way to increase carbon sequestration capacity of forest regions, because forest owners plant normally fast-growing tree species (*Reyes, León & Herrero, 2019*).

*Pinus durangensis* and *P. cooperi* are species frequently used in forest reforestations in the state of Durango (*Prieto et al., 2016*). However, successful forest plantation requires planting quality seedlings with optimal growth potential. Thus, foresters need to use seedlings with plant attributes (i.e., shoot height, stem diameter, root mass, shoot to root ratio, drought resistance, mineral nutrient status) that favor the best chance of successful establishment once they are field planted (*Grossnickle, 2012*; *Pérez-Luna et al., 2019*; *Pérez-Luna et al., 2020*). The use of quality seedlings favors success in plantations under climate change conditions (*Vallejo et al., 2012*). In Mexico 57% of mortality is caused by the poor quality of the plant (*Prieto et al., 2016*), also another important factor are the deficiencies that occur during the planting process (*Burney et al., 2015*), for example, if the strain becomes too deep the plant will suffocate and if the strain becomes shallow, the roots could be left on the surface and the plant will dehydrate, in addition, covering the stem with too much soil reduces the vigor of the plant and makes difficult its access to water (*CONAFOR, 2010*).

For the proper management of forests in Mexico, including areas planted with the main commercial tree species, different silvicultural systems are employed in order to ensure the regeneration of the site: The Mexican Irregular Forest Management Method (MMOBI) and the Silvicultural Development Method (MDS) (*Solís et al., 2006*; *Pérez-Verdín et al., 2009*; *Pérez-Rodríguez et al., 2013*). The former is used in forest stands with a high tree species richness, and the latter in stands dominated by one or two tree species of pine (*Hernández-Díaz et al., 2008*). Moreover, in areas covered with little or no slope, the use of a clear-cutting as a regeneration method can be also used. It is characterized by having periodic crops, determined by commercial rotations (*Gadow, Sanchez & Aguirre, 2004*), and their regeneration can be natural or artificial. If the regeneration is artificial, then the use of fast-growing species is preferable.

Although clear-cutting can be used successfully in Mexican forests, this silvicultural method is still considered inappropriate, because complete tree removal of an area may cause degradation of forest components such as soil, water quality, fauna, etc., at least at the beginning of the forestry process (*Keenan & Kimmins, 1993*; *Hernández, Jaeger & Samperio, 2017*; *Monárrez-González et al., 2018*). However, its use has economic advantages. On the one hand, an intermediate economic income is possible, through silvicultural interventions (thinnings); on the other, the reduction of competition improves the dimensions of the trees that remain standing and the $CO_2$ sequestration increases (*Rodríguez-Larramendi et al., 2016*; *Rodríguez-Ortiz et al., 2019*). Investigations that address the survival behavior and the carbon content of mixed *Pinus cooperi* and *P. durangensis* plantations in the initial phase are scarce (*Návar et al., 2004*). The study of these aspects is important for the proper management and conservation of forest plantations in Mexico. As a hypothesis of this study, we consider that plantations of *Pinus cooperi* and *P. durangensis* established after clear-cuttings, show high levels of survival and growth, and contribute efficiently to carbon sequestration. Thus, the objectives of this work were to evaluate survival, the width of tree rings and carbon content in a plantation of *Pinus durangensis* and *P. cooperi* at a site exposed to clear-cutting in the State of Durango, Mexico.

## MATERIALS & METHODS

### Study area

This study was conducted in the private property "Las Veredas" in the municipality of San Dimas, Durango, Mexico, which belongs to the Compañía Silvícola Chapultepec, S. DE R. L. DE C. V., in the coordinates of the site are: 24°20′40″N and 105°51′20″W (Fig. 1). Compañía Silvícola Chapultepec S. DE R.L. de C.V. provided a field permit for access to the study area for data collection. The climate is temperate with a brief rainy season during the summer months ($C_W$), and the temperature ranges between −3 and 18 °C (*García, 2004*). The topography is characterized by hills, with slopes ranging from soft to medium (0 to 50%). The area occupied by the clear-cutting has a slope of 9% and an altitude of 2803 m asl. The average annual rainfall recorded during the period 2010 to 2018 was 1,034.5 mm (this value was recorded by the weather station in the town of Vencedores, located 15 km away from the study area). Rainfall occurs in the months of June, July, August and September; The first frost occurs in October and the last frost occurs in June; snowfall occurs most frequently in the months of December and January (*FSC & SmartWood, 2002*).

The vegetation is characterized by mixed coniferous and broadleaved forests. The dominant pine species are: *Pinus durangensis* Martínez, *P. cooperi* Blanco, *P. teocote* Schl. and *P. strobiformis* Engelm. The main oak species are: *Quercus rugosa* Née and *Q. sideroxyla* Bonpl, in addition some species of the genera *Juniperus, Arbutus,* and *Alnus* are also part of the forest composition (*González-Elizondo et al., 2012*).

### Forestry background

Before the clear-cutting, the stand had a stocking of 220.4 m³ ha⁻¹ of trees of the following genera: *Pinus, Quercus, Juniperus* and *Arbutus* (*PMF, 2010*). The trees were harvested at

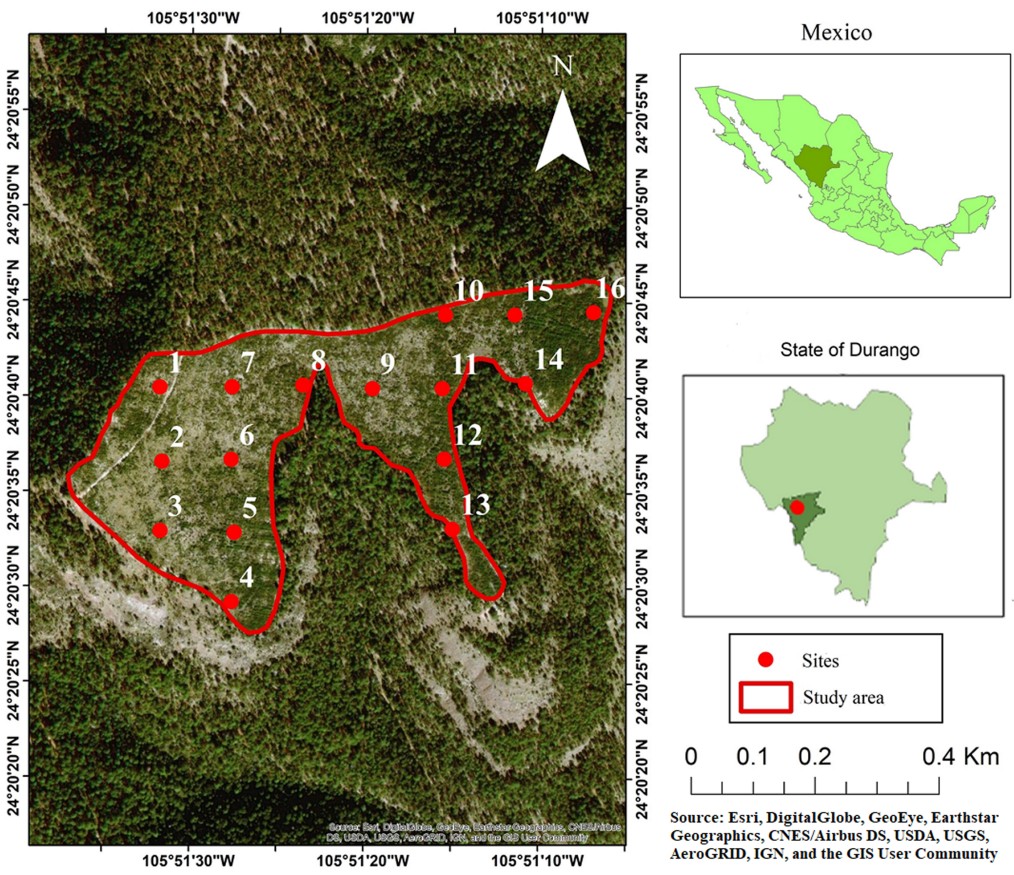

**Figure 1** **Study area, showing the locations of the sampling sites used in the analysis.** Photo credit: Esri, DigitalGlobe, GeoEye, Earthstar Geographics, CNES/Airbus DS, USDA, USGS, AeroGRID, IGN, and the GIS User Community.

the beginning of 2010 on an area of 21.40 ha. Months later, during the rainy season, the plantation was established with seedlings of *P. durangensis* and *P. cooperi* (it was not possible to know the proportion, but *P. durangensis* was planted in greater quantity). The plantation was produced with germplasm collected from trees growing in natural stands next to the study area. Land preparation consisted of clearing, scattering and applying the controlled burning of forest waste. To improve soil conditions, the ground was plowed using a D-6 track-type tractor, equipped with a ripper, breaking equidistant lines (2 m) at a depth of 60 cm, in the perpendicular direction to the slope. The seedlings were 12 months old with a height of between 15 and 20 cm and a diameter at the base of the stem of 5 mm. They were planted using the common strain method in a "real frame" at a spacing of 2 × 2 m to generate a density of 2,500 seedlings ha$^{-1}$.

## Field evaluation

Sixteen (16) circular sites of 100 m$^2$ were established, distributed completely randomly to evaluate survival according to the recommendations of *CONAFOR (2013)*. In addition to survival, variables such as diameter (cm) and height (m), were measured using a Vernier
and hypsometer (Vertex V®), respectively. The sample implied the demolition of 32 trees, distributed proportionally to the diametric categories (16 of each tree species). Two cross-sections were obtained from each tree (over-bark and under-bark): one was cut at the base of the tree and the second one at 1.3 m above ground.

The cross-sections were labeled, dried and polished, and the measurement of each growth ring was subsequently performed using a stereomicroscope with precision in μm. This process was carried out in the dendroecology laboratory at the Facultad de Ciencias Forestales of the Universidad Juárez del Estado de Durango. The year-by-year width of each tree ring was estimated by the average of four-way measurements. The measurements were performed by starting from the year 2017, looking back to previous years.

## Estimation of biomass and carbon content

In order to obtain the total aerial biomass, the equation developed by *Návar et al. (2004)* was used for the species in question, which takes the basal diameter as a predictive variable (Eq. (1)).

$$y_i = a(DB)^b. \tag{1}$$

*Where*
$y_i$ = Total aerial biomass
$a = 0.0199y$  $b = 2.5488$
$DB$ = Basal tree diameter (cm).

The carbon content for *Pinus cooperi* was calculated according to the percentage reported by *Pompa-García et al. (2017)*, who indicate that, the carbon concentration for this species is 49.64% of the total aerial biomass. For *Pinus durangensis* the carbon concentration reported for (*Hernández-Vera et al., 2017*) was used (50.36% of the total aerial biomass).

Considering that the terrain is relatively flat and homogeneous, an analysis of variance (ANOVA) was performed under a completely randomized experimental design for the statistical analysis of the data. The classification variables were the two studied tree species and the tree age. The response variables were the year-by-year width of the tree ring, the biomass and the carbon content. The Shapiro–Wilk Normality Test was used to evaluate data normality and equality of variances for all response variables. Significant differences ($p \leq 0.05$) for the year-by-year width of the tree rings were evaluated with the Tukey Means Comparison Test, while the Kruskal–Wallis Non-Parametric Test was used to evaluate significant differences among the median values of biomass and carbon content because these two variables did not meet the normality data assumption. These analyzes were performed using R® statistical software (*R Core Team, 2019*).

## RESULTS

Table 1 shows the descriptive statistics of the sampled trees. Similarity is observed in the parameters evaluated (diameter at the base and total height), which means that, at the age of eight years, the studied tree species show similar growth patterns.

Eight years after the establishment of the plantation, an average survival of 75.2% was observed for the two species, which corresponds to a density of 1,881 trees per ha (Table 2).
**Table 1  Descriptive statistics for the diameter at the tree base and height of the sampled trees of *Pinus durangensis* and *P. cooperi* that were analyzed in this study (eight years old).**

| Species | Trees | DB (cm) | | | | Height (m) | | | |
|---|---|---|---|---|---|---|---|---|---|
| | | M | Var. | Std | C.V. | M | Var | Std | C.V. |
| *P. durangensis* | 16 | 8.21 | 1.13 | 1.06 | 0.13 | 3.47 | 0.33 | 0.58 | 0.17 |
| *P. cooperi* | 16 | 8.17 | 2.24 | 1.5 | 0.18 | 3.3 | 0.67 | 0.82 | 0.25 |

Notes.

DB, basal diameter; M, Mean; Var, Variance; Std, Standard deviation; C.V., Coefficient of Variation.

**Table 2  Number of trees observed by site and tree species.**

| Site | n | N | % | *Pinus durangensis* | | *P. cooperi* | |
|---|---|---|---|---|---|---|---|
| | | | | n | % | n | % |
| 1 | 19 | 1,900 | 76 | 14 | 56 | 5 | 20 |
| 2 | 10 | 1,000 | 40 | 7 | 28 | 3 | 12 |
| 3 | 18 | 1,800 | 72 | 11 | 44 | 7 | 28 |
| 4 | 20 | 2,000 | 80 | 12 | 48 | 8 | 32 |
| 5 | 15 | 1,500 | 60 | 7 | 28 | 8 | 32 |
| 6 | 18 | 1,800 | 72 | 13 | 52 | 5 | 20 |
| 7 | 19 | 1,900 | 76 | 13 | 52 | 6 | 24 |
| 8 | 21 | 2,100 | 84 | 12 | 48 | 9 | 36 |
| 9 | 20 | 2,000 | 80 | 8 | 32 | 12 | 48 |
| 10 | 24 | 2,400 | 96 | 13 | 52 | 11 | 44 |
| 11 | 23 | 2,300 | 92 | 16 | 64 | 7 | 28 |
| 12 | 23 | 2,300 | 92 | 21 | 84 | 2 | 8 |
| 13 | 18 | 1,800 | 72 | 12 | 48 | 6 | 24 |
| 14 | 11 | 1,100 | 44 | 8 | 32 | 3 | 12 |
| 15 | 23 | 2,300 | 92 | 18 | 72 | 5 | 20 |
| 16 | 19 | 1,900 | 76 | 15 | 60 | 4 | 16 |
| Mean | 18.81 | 1,881.25 | 75.20 | 12.5 | 50 | 6.31 | 25.25 |

Notes.

$n$, number of trees recorded in site $i$; $N$, Number of trees per hectare; %, Percentage of survival.

From studied sites stand out 10 and 2 which showed the highest and lowest survival rates, respectively. No survival comparisons were made, because it was not possible to know the proportion in which both species were planted, however, *Pinus durangensis* was found to be planted in a greater quantity (almost double according to the observed average of the number of trees of the sampling sites), and assuming that survival had been the same for the two studied trees species, an initial density of 2,500 trees ha$^{-1}$ was planted (25 trees in 1,000 m$^2$).

The analysis of variance indicated that there are highly significant statistical differences ($p < 0.001$) in the year-by-year width of the tree rings of the studied tree species. The Tukey test (Fig. 2), forms 6 groups for both tree species, among these, the values of the first and fifth year stand out, which were, respectively, the minor and major width rings. Starting from fifth year, the width of the tree rings shows a decline, and allows their values to be grouped with the values of the third and fourth years. Although the width of the tree

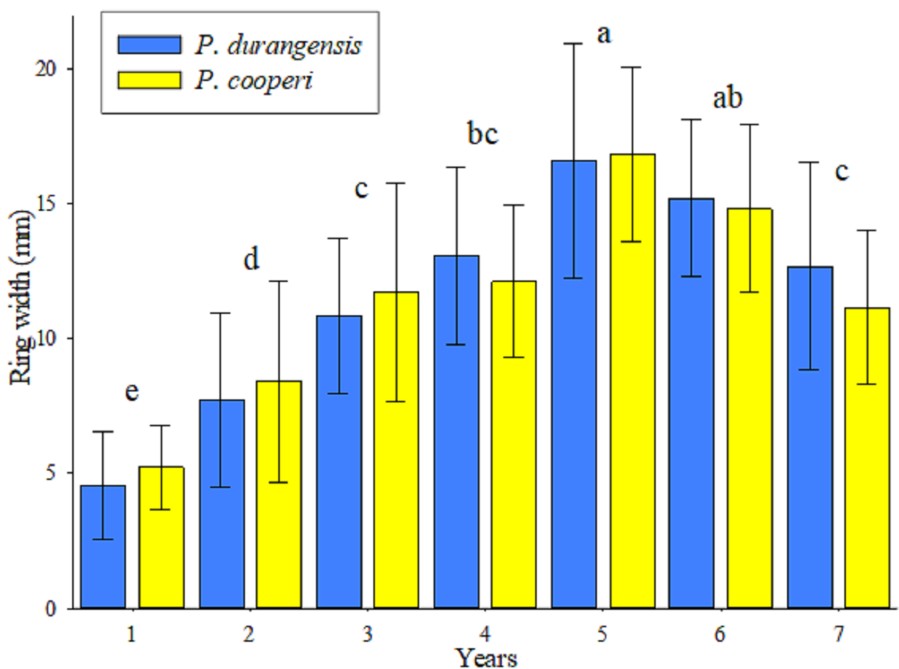

**Figure 2** Average values (bars), standard deviation (lines) and groups of means according to Tukey (letters) of the ring width by species and year.

rings was slightly larger in *P. cooperi*, no significant statistical differences ($p = 0.9336$) were found among the two studied tree species. The upper and lower horizontal lines crossing the vertical bars (standard deviation) indicate that the year-by-year width growth of the tree rings has been similar among the tree species until the period of time evaluated in this study.

The Kruskal–Wallis Test indicated that there are significant differences in the estimates of year-by-year biomass and carbon content ($p \leq 0.05$). The comparison of medians using the Bonferroni method does not show significant differences in those years whose observations were consecutive. In contrast, comparisons between discontinuous years showed significant differences ($p < 0.01$) (Table 3).

Estimates of year-by-year biomass and carbon content among tree species did not show significant statistical differences ($p \geq 0.05$).

The accumulation of biomass showed higher values after the fifth year in the studied species, which is a product of the increase in the dimensions of the BD of the trees (Fig. 3).

Estimates of biomass and carbon content year-by-year of the studied species are shown in Table 4. It is observed that the accumulation of biomass and carbon content increases with increasing age. However, *Pinus cooperi* shows a decrease in the seventh year with respect to the previous year. Considering that a tree has in average an estimate of 4.42 and 2.21 kg of biomass and carbon content, respectively, at the age of 7 years, the studied plantation accounts for 8,315.12 and 4,157.56 kg ha$^{-1}$ of aerial biomass and carbon, respectively.
**Table 3** Pairwise comparisons of the estimation of year-by-year biomass and carbon content according to the Bonferroni medians comparison test.

| Comparison (year) | Biomass | Carbon |
|---|---|---|
| 1 vs 2 | 0.9507 | 0.9508 |
| 1 vs 3 | 0.0013[**] | 0.0014[**] |
| 2 vs 3 | 0.9816 | 0.9907 |
| 1 vs 4 | <0.0001[***] | <0.0001[***] |
| 2 vs 4 | 0.0034[**] | 0.0035[**] |
| 3 vs 4 | 1 | 1 |
| 1 vs 5 | <0.0001[***] | <0.0001[***] |
| 2 vs 5 | <0.0001[***] | <0.0001[***] |
| 3 vs 5 | <0.0001[***] | <0.0001[***] |
| 4 vs 5 | 0.1064 | 0.1065 |
| 1 vs 6 | <0.0001[***] | <0.0001[***] |
| 2 vs 6 | <0.0001[***] | <0.0001[***] |
| 3 vs 6 | <0.0001[***] | <0.0001[***] |
| 4 vs 6 | <0.0001[***] | <0.0001[***] |
| 5 vs 6 | 1 | 1 |
| 1 vs 7 | <0.0001[***] | <0.0001[***] |
| 2 vs 7 | <0.0001[***] | <0.0001[***] |
| 3 vs 7 | <0.0001[***] | <0.0001[***] |
| 4 vs 7 | <0.0001[***] | <0.0001[***] |
| 5 vs 7 | 1 | 1 |
| 6 vs 7 | 1 | 1 |

**Notes.**
[*] A significant difference at $p < 0.05$.
[**] A significant difference at $p < 0.01$.
[***] A significant difference at $p < 0.001$.

## DISCUSSION

The results of survival reported in this work (75.2%) are superior than the findings reported by *Prieto et al. (2016)*, who found out an average of 43% in a forest plantation studied in Durango State. These survival results are also superior to those showed by *Bojórquez, Rodríguez & Flores (2015)* in 25-year-old plantations of *P. durangensis*, *P. engelmannii*, *P. cooperi* and *P. arizonica* var. in Durango State. Although different species were dealt with in a similar study, our results are also better than those reported by *Vásquez-García et al. (2016)*, who informed an average survival of 69% in four-year old plantations of *P. greggii* Englem and *P. oaxacana* Mirov—in three communities in the High Oaxacan Mixtec region in Southern Mexico. However, our survival results were inferior than those reported by *Prieto et al. (2007)* in an 18-month *Pinus cooperi* plantation established on a site in the municipality of Durango, reaching a survival rate of 85.6%. They are also inferior than those informed by *Prieto et al. (2018)*, who reported a 93% of survival in a *Pinus cooperi* plantation after 13 months of its establishment at a site in Agua Zarca, Otinapa, Durango.

*Prieto et al. (2018)*, argue that successful plantations and reforestations depend on the quality of the plant, as it is decisive for its adaptation and development after planting
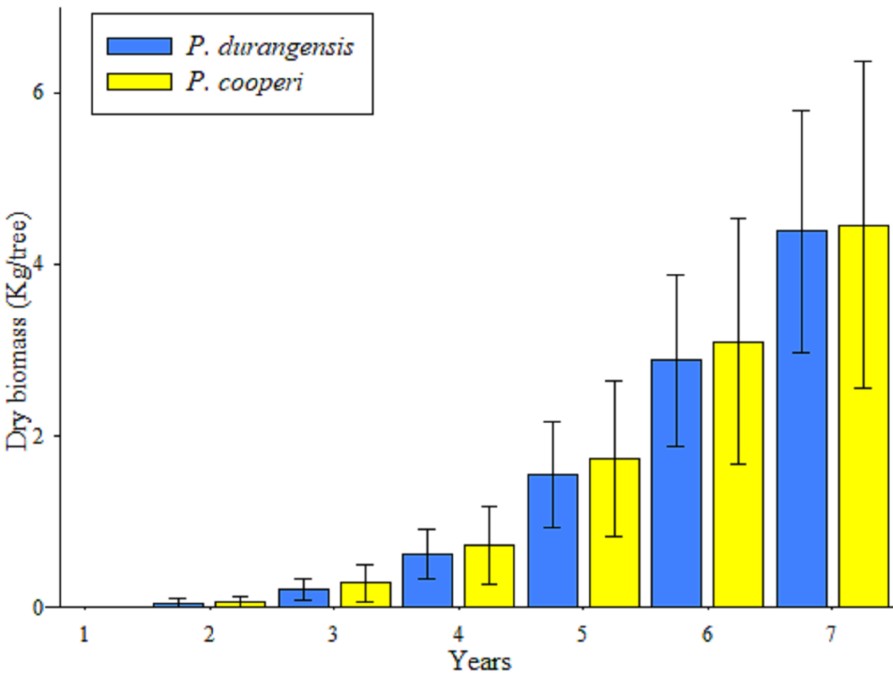

**Figure 3** Average values and standard deviation of the total accumulated aerial biomass by tree of *P. durangensis* and *P. cooperi*.

**Table 4** Year-by-year biomass and carbon content for the studied pine tree species planted in a site after being harvested with a clear-cutting.

| Age (year) | Biomass *P. durangensis* (kg Tree$^{-1}$) | | Biomass *P. cooperi* (kg Tree$^{-1}$) | | Carbon *P. durangensis* (kg Tree$^{-1}$) | | Carbon *P. cooperi* (kg Tree$^{-1}$) | |
|---|---|---|---|---|---|---|---|---|
| | Mean | Std | Mean | Std | Mean | Std | Mean | Std |
| 1 | 0.006 | 0.005 | 0.008 | 0.005 | 0.003 | 0.003 | 0.004 | 0.002 |
| 2 | 0.047 | 0.045 | 0.060 | 0.049 | 0.024 | 0.023 | 0.030 | 0.024 |
| 3 | 0.164 | 0.086 | 0.223 | 0.166 | 0.082 | 0.043 | 0.111 | 0.083 |
| 4 | 0.404 | 0.183 | 0.429 | 0.250 | 0.203 | 0.092 | 0.213 | 0.124 |
| 5 | 0.934 | 0.397 | 1.014 | 0.470 | 0.470 | 0.200 | 0.503 | 0.233 |
| 6 | 1.324 | 0.451 | 1.366 | 0.583 | 0.667 | 0.227 | 0.678 | 0.290 |
| 7 | 1.509 | 0.608 | 1.362 | 0.559 | 0.760 | 0.306 | 0.676 | 0.278 |
| Total | 4.388 | 1.776 | 4.461 | 2.083 | 2.210 | 0.894 | 2.215 | 1.034 |

**Notes.**
Std, Standard deviation.

(*Fontana, Pérez & Luna, 2018*). In our study, it is observed that both species show uniform growth from their establishment to the fifth year of planting. However, from the sixth year onwards, plant growth tends to decline, remaining low until the end of the study, being more noticeable in *P. cooperi*. *Crecente-Campo et al. (2007)* documented that the decrease in plant growth is influenced among other factor by the competition that trees experience from other trees, and that competition increases with increasing age of the plantation (*Soto-Cervantes et al., 2016*).

Although this work does not include an analysis that correlates ecological factors with tree growth, studies performed by *Anchukaitis et al. (2013)* and *Gutiérrez-García & Ricker (2019)*, documented that maximum temperature negatively influences radial growth of the trees. However, the growth, and consequently, the $CO_2$ sequestration may vary with environmental conditions of the site and the tree species evaluated (*Haghshenas et al., 2016*; *Pompa-García & Sígala-Rodríguez, 2017*; *Lanza, Chartier & Marcora, 2018*). These growth patterns can be attributed to the fact that, at the beginning of the plantation, the genotype-environment interaction and the plasticity of the species influence the adaptability of the plants to a specific sites (*Alía, 2006*; *Thomas et al., 2015*).

The width of the tree rings is the most used parameter to evaluate the growth rate of the trees (*Dobner, Huss & Tomazello, 2018*). In this study, the lowest width ring growth value was presented during the first year, this can be attributed to plant adaptation to transplant stress (e.g., drought, salinity, and temperature extremes) (*Pérez-Luna et al., 2020*). On the other hand, the tree development during the fifth year was higher and after that it showed some reduction in growth caused by tree competition (*Crecente-Campo et al., 2007*). The average width growth of the rings observed for both tree species was 11.51 mm $yr^{-1}$. *Pompa-García et al. (2018)*, evaluated the width growth rings in *P. arizonica* y *P. cembroides* Zucc in northern Mexico and found that their average increase was 1.84 mm $yr^{-1}$ and 1.73 mm $yr^{-1}$, respectively.

A study in *Pinus taeda* L. developed by *Dobner, Huss & Tomazello (2018)*, found out that the width growth of tree rings varied between 6 to 9 mm during the first three to six years before performing silvicultural activities. On the other hand, *Melandri, Dezzeo & Espinoza de Pernía (2007)* in a study with *Pinus caribaea* var. hondurensis in Venezuela indicated that the width growth of tree rings is dependent on various site factors, as well as atmospheric, in this case they pointed out that different rainfall regimes yielded significantly differences in with growth rings.

Several studies related the radial growth rate and the characteristics of the rings with environmental variables such as temperature, precipitation and light intensity (*García-Suárez, Butler & Baillie, 2009*; *Chacón-de la Cruz & Pompa-García, 2015*). However, due to the dynamics of the exchange activity during the formation of the wood, the relationship of growth with climatic conditions is very complex and inconstant (*Zywiec et al., 2017*).

Studies indicate that tree spacing influences total tree volume, width growth of rings, and length and diameter of branches (*Hart, 2010*). Also, with increasing thinning intensity the effect on the thickness of the width of tree rings is greater (*Zhang et al., 2006*). *Baldwin et al. (2000)*, studied the effects of spacing and thinning on stand and tree characteristics of a 38-year-old *Pinus taeda* L plantation and found out that, at higher initial spacing, trees develop greater diameter and crown length. This favors the formation of wider tree rings and more early wood (*Rossi, Morin & Deslauriers, 2012*). *Rodríguez-Ortiz (2010)*, recommended an initial spacing of 2.40 to 2.75 m for *Pinus patula* Schl, because it is a tree species sensitive to competition during its first years of development.

In this study, estimates of year-by-year biomass and carbon content were analyzed for individual trees, however, estimates per hectare were not performed because survival in previous years was unknown and results extrapolation based on survival at the time of

assessment could be considered erroneous. In this study the average carbon content per tree was estimated in 0.30 kg yr$^{-1}$ for both studied tree species. *Pompa-García et al. (2018)* estimated that adult trees of *Pinus arizonica* and *P. cembroides* accumulate 4.80 kg C yr$^{-1}$ and 4.84 kg C yr$^{-1}$, respectively. *Pacheco et al. (2007)* evaluated the accumulated biomass content in a six-year-old *Pinus greggii* plantation, and found out that the average aerial dry biomass was 8.0 kg per tree. These results are above those observed in this study with an average per tree of 4.39 kg for *Pinus durangensis* and 4.46 kg for *P. cooperi*.

Intensive forestry is key to maintaining or increasing stand productivity in the future, wood biomass from plantations of fast-growing trees is an alternative because it is produced in short periods of time (*Thiers, Gerdinga & Schlatter, 2007*). However, intensive forest management may impact the conservation of plant diversity and the regulation of water flows (*Monárrez-González et al., 2018*). The results of this research suggest that clear-cuttings are suitable logging practices for the studied tree species in the study area, especially because they create optimal growing conditions for pine trees. However, after opening the gaps it is recommended to replant the area with certain additional complementary practices to protect and guarantee its good development (e.g., fencing, firebreaks, pest prevention, etc.), and avoid risk of soil erosion and loss of biodiversity. The estimates of carbon content made in this study coincide with the results reported by *Pacheco et al. (2007)*, in the sense that young plantations have high growth rates and therefore they also have a greater potential for carbon sequestration.

## CONCLUSIONS

According to the results of the present study, the observed survival percentage (75.2%) is considered high in comparison with other similar studies. It indicates that both *Pinus durangensis* and *P. cooperi* adapt and develop well in areas harvested with clear-cutting in the forests of Durango. The width growth of the tree rings was similar in the two species but different among the years evaluated. No significant differences were found in terms of biomass and carbon content among the studied tree species. The accumulation of biomass and carbon content observed in the studied forest plantation is considered to be high, accounting for 8,315.12 and 4,157.56 kg ha$^{-1}$, respectively, at the age of 7 years. The study reveals that clear-cuttings can be successfully used as logging practices in the study area to create even aged pine stands and increase the productivity of these forests in terms of timber and $CO_2$ sequestration, among other ecosystems services.

## ACKNOWLEDGEMENTS

Our thanks to C.P. Alfonso Gerardo Fernández de Castro Toulet, legal representative of the Compañía Silvícola Chapultepec S. DE R.L. de C.V., for providing access to the study area for data collection; to UCODEFO No. 4 (Unidad de Conservación y Desarrollo Forestal No. 4 La Victoria Miravalles SC), for providing information related to the study area; we also thank DendroRed (https://dendrored.ujed.mx).

### Funding

This work was supported by Consejo Nacional de Ciencia y Tecnología (CONACYT-No. 297160 and A1-S-21471). The funders had no role in study design, data collection and analysis, decision to publish, or preparation of the manuscript.

### Grant Disclosures

The following grant information was disclosed by the authors:
Consejo Nacional de Ciencia y Tecnología: CONACYT-No. 297160.

### Competing Interests

The authors declare there are no competing interests.

### Author Contributions

- Jesús Alejandro Soto-Cervantes and Pedro Antonio Dominguez-Calleros conceived and designed the experiments, performed the experiments, analyzed the data, prepared figures and/or tables, authored or reviewed drafts of the paper, and approved the final draft.
- Artemio Carrillo-Parra conceived and designed the experiments, analyzed the data, prepared figures and/or tables, authored or reviewed drafts of the paper, and approved the final draft.
- Rodrigo Rodríguez-Laguna analyzed the data, prepared figures and/or tables, authored or reviewed drafts of the paper, and approved the final draft.
- José Javier Corral-Rivas conceived and designed the experiments, performed the experiments, analyzed the data, authored or reviewed drafts of the paper, and approved the final draft.
- Marín Pompa-García performed the experiments, analyzed the data, authored or reviewed drafts of the paper, and approved the final draft.

### Field Study Permissions

The following information was supplied relating to field study approvals (i.e., approving body and any reference numbers):

Field experiments were authorized by the C.P. Alfonso Gerardo Fernández de Castro Toulet, General Manager and legal representative of the Compañía Silvícola Chapultepec S. DE R. L. DE C.V.

### Data Availability

The raw data are available as Supplemental File.

### Supplemental Information

Supplemental information for this article can be found online at http://dx.doi.org/10.7717/peerj.9506#supplemental-information.

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
