# Peer review of "Survival, growth and carbon content in a forest plantation established after a clear-cutting in Durango, Mexico"

_PeerJ, doi:10.7717/peerj.9506_

## Round 0.1 · original submission · Major Revisions

Two reviewers have recognized the potential value of your manuscript and recommended that it be accepted after further revision. Please consider all of their suggestions and document your responses to each suggestion, specifying how your revision addresses those suggestions or why you chose not to follow the reviewer's guidance. Please pay special attention to the first reviewer's suggestions for making stronger inferences about carbon sequestration based on additional calculations. Both reviewers also highlighted that the Introduction and Discussion should be improved by considering additional and/or alternative sources of information and hypotheses.

·

Basic reporting

The English used presents some failures, but which are salvageable, when correcting the authors. Literature references are insufficient, especially in the discussion section, it also presents author's writing failures, years and format in general. If the authors pay special attention to this section, the manuscript can be improved. The article has professional structure and the results are correlated with the hypotheses.

Experimental design

The work is original and is necessary for many species in Mexico.The methodological part requires specifying some details of the experimental design and other characteristics indicated in the writing.

Validity of the findings

In addition to the comments made in the original text of the article, the potential for carbon sequestration of both species would be lacking. The authors can achieve this by incorporating the accumulation rates of biomass-C, based on the amplitude of the growth rings and providing these values with reference to the hectare. This so that part of the conclusion is not speculated and the payment of environmental services is magnified by this concept.

Additional comments

Authors should be careful in the journal format, as the manuscript must be substantially improved.

·

Basic reporting

The paper addresses the role of reforestation (forest plantation) in carbon sequestration or climate change mitigation, which is an important subject in forestry. However, the authors did not completely develop the research insights in the Introduction, rather they included not relevant information (see specific comments below). They should state clearly the research question, for example, are there differences in the contribution of individual species in carbon sequestration? Is it reasonable to expect any synergistic response of both species together in the reforestations? In this sense the Introduction section is not complete.
The hypothesis is rather general and it was not fully proved since their findings does not suggests that reforestation, by itself, are efficient way of mitigating climate change or increasing carbon sequestration, because species response varies with site conditions, seedling history, etc.

Experimental design

Not clearly stated in terms of trataments effect. Please provide research questions and an explanation on effects tested in ANOVA.

For data analysis (Anova) the authors should state clearly what treatments effects are testing for and why. For example, why comparisons among species was carried out for most variables but not for survival? The authors could improve the Discussion section considering that Pinus cooperi survival was clearly lower than P. durangensis, since this might have important implications for results in all other variables.

Validity of the findings

I suggest to restate the Conclussions since most statements are not based on the study results. For example, total survival percentages does not show species performance.

---

## Round 0.2 · Minor Revisions

I think the manuscript has been improved, but that the Introduction should be further expanded and revised. This might also require minor corresponding revisions to the Discussion. Please see my comments in the attached annotated document, and submit your responses to these comments with the further revised manuscript.

---

## Round 0.3 · accepted · Accept

The manuscript has been revised sufficiently. I appreciate the authors' continued effort to improve the manuscript.